# Damage to the Testicular Structure of Rats by Acute Oral Exposure of Cadmium

**DOI:** 10.3390/ijerph18116038

**Published:** 2021-06-04

**Authors:** Tariq Iqbal, Maosheng Cao, Zijiao Zhao, Yun Zhao, Lu Chen, Tong Chen, Chunjin Li, Xu Zhou

**Affiliations:** College of Animal Sciences, Jilin University, Changchun 130013, China; iqbaltariq9917@mails.jlu.edu.cn (T.I.); caoms18@mails.jlu.edu.cn (M.C.); zhaozj18@mails.jlu.edu.cn (Z.Z.); zhao_yun@jlu.edu.cn (Y.Z.); luchen@jlu.edu.cn (L.C.); ctokay@163.com (T.C.)

**Keywords:** spermatogenesis, cadmium, DNA damage, daily sperm production, industrial toxicology

## Abstract

Cadmium (Cd) is one of the most important heavy metal toxicants, used throughout the world at the industrial level. It affects humans through environmental and occupational exposure and animals through the environment. The most severe effects of oral exposure to Cd on the male reproductive system, particularly spermatogenesis, have not been discussed. In this study, we observed the damage to the testes and heritable DNA caused by oral exposure to Cd. Adult male Sprague–Dawley rats were divided into four groups: a control group and three groups treated with 5, 10, and 15 mg Cd/kg/day for 17 days by oral gavage. Our results revealed that Cd significantly decreases weight gain in 10 and 15 mg/kg groups, whereas the 5 mg/kg groups showed no difference in weight gain. The histopathology showed adverse structural effects on the rat testis by significantly reducing the thickness of the tunica albuginea, the diameter of the tubular lumen, and the interstitial space among seminiferous tubules and increasing the height of the epithelium and the diameter of the seminiferous tubules in Cd treated groups. Comet assay in epididymal sperms demonstrated a significant difference in the lengths of the head and comet in all the 3 Cd treated groups, indicating damage in heritable DNA, although variations in daily sperm production were not significant. Only a slight decrease in sperm count was reported in Cd-treated groups as compared to the control group, whereas the tail length, percentage of DNA in head, and tail showed no significant difference in control and all the experimental groups. Overall, our findings indicate that Cd toxicity must be controlled using natural sources, such as herbal medicine or bioremediation, with non-edible plants, because it could considerably affect heritable DNA and induce damage to the reproductive system.

## 1. Introduction

Demand for different products has increased as a result of the massive growth of the human population, and different industries have been established to meet these demands; unfortunately, these industries have brought about increased mining and industrial processing activities, resulting in environmental pollution [1,2]. Several life-threatening pollutants are found in the soil, water, and air [3]. Among these pollutants, Cadmium (Cd) is considered one of the most lethal heavy metal toxicants because of its hazardous properties, for example, severe toxicity, global availability, transferability, and persistence [4]. According to the Agency for Toxic Substances and Disease Registry (ATSDR), Cd is the sixth most hazardous chemical for living organisms [5]. The World Health Organization reported that, along with natural activities (e.g., volcanic eruption, river transport, weathering and erosion), the concentration of Cd in the environment increases with human activities, such as combustion of fossil fuel, mining, smelting, and refining of non-ferrous metals, smoking of tobacco, production of phosphate fertilizers, incineration of municipal wastes containing Cd batteries and plastics, and recycling of Cd-plated scrap [6,7,8]. Jarup and Akesson (2009) reported that increased mining and industrial activities and the use of Cd-containing fertilizers result in the contamination of soil and absorption of large amounts of Cd by plants grown for human and animal consumption [9]. In most parts of the world, the diet has been reported as the chief source of environmental Cd exposure in non-smokers [9,10,11].

The concentration of Cd in the diet varies significantly. Common contributors of Cd in humans are fiber-rich diets, such as vegetables, shellfish, and cereals; in some areas, rice is reported to be a common source of Cd [12,13]. Another major cause of Cd exposure is tobacco smoking. A cigarette usually contains approximately 1–2 µg of Cd, depending on the brand; approximately 10% of this Cd is inhaled, and an estimated 50% of inhaled Cd is absorbed by the lungs [14,15]. The average Cd consumption from the diet usually varies from 8 µg to 25 µg per day [9,16,17,18,19,20,21,22,23]. This consumption may be even higher in some parts of the world (e.g., Japan) [9].

Studies have measured the damage caused by Cd to the body by using various routes of administration because the distribution and absorption of different elements in food are affected by the administration route [9,24]. The quantity of Cd absorbed by the body is higher when the metal is administered intraperitoneally (i.e., i.p. injection) than when it is administered orally, and through inhalation of cigarette smoke or occupational exposure to fumes containing high concentrations of Cd (ATSDR 2008) [23,25,26]. In daily life, exposure to Cd commonly occurs through food sources; therefore, knowledge of the damage caused by Cd, when it is absorbed in the intestine, is important. Moreover, the effect of Cd exposure on heritable DNA has rarely been discussed. The present study aimed to observe the microscopic damage brought about by Cd on the testis and heritable DNA of Sprague–Dawley (SD) rats.

## 2. Materials and Methods

### 2.1. Animals

Adult male SD rats (age, 70–85 days) were obtained from the Animal Facility of College of Animal Sciences, Jilin University, Changchun, China, and kept in plastic cages with a stainless steel top at a controlled temperature of 24 ± 2 °C and 50–60% humidity for 1 week to acclimatize to the lab environment. All the rats were maintained at a 12 h/12 h light/dark cycle and fed with standard laboratory food. Tap water was made available ad libitum. The animal handling, treatment, and sacrifice protocols were approved by the College of Animal Sciences, Jilin University China (Permit Number SY201909012).

### 2.2. Experimental Design

Twenty-four adult male SD rats were divided into four groups of six animals each. The first group served as the control group and was given 1.5 mL of saline via a feeding tube. The remaining three groups were treated with 5, 10, and 15 mg Cd/kg/day in the form of cadmium Chloride (CdCl_2_) (Tianjin Guangfu Technology Development Co., Ltd; Tianjin, China) solution with a feeding tube. In brief, a stock solution of 0.1 M CdCl_2_ was prepared in double-distilled water and then a required quantity of Cd from this stock solution was mixed with saline in accordance with the weight of the animals to obtain a total solution volume of 1.5 mL for each animal.

The three doses of Cd were selected based on the results of previous studies. According to ATSDR, the lethal dose of Cd with a 50% kill rate (LD_50_) is 100–300 mg Cd/kg [27,28,29,30,31]. The current doses (i.e., 5, 10, and 15 mg Cd/kg/day) were selected because the smallest possible lethal oral dose of Cd is 15.3 mg/kg/day. All our doses were under that range, and these doses did not kill the animals examined in previous studies [32,33].

### 2.3. Experimental Duration (Animal Trails)

All of the treatments in our experiment were based on data from the ATSDR, which divides Cd exposure into three categories on the basis of health effects: acute (i.e., exposure for 2 weeks or less), intermediate (i.e., exposure for 15 days to 1 year), and chronic (i.e., exposure for 365 days or more) [34]. Cilenk (2016) and his team studied cadmium toxicity by using an i.p. injection for 17 days [34]. In daily life, Cd exposure usually occurs through the diet; thus, all the doses in the current experiment were administered orally ((i.e., oral gavage) for 17 days, including weekends. On day 18, roughly 24 h after administration of the last dosage, the animals were anesthetized with 750 mg/kg of i.p. injection of 2,2,2-tribromoethanol solution and then sacrificed according to the guidelines of Jilin University (https://sydw.jlu.edu.cn/info/1009/2196.htm (accessed on 4 May 2021); http://202.198.25.15/uhtbin/cgisirsi/x/0/0/5?searchdata1=548824{ckey}; (accessed on 4 May 2021); https://www.lac.pku.edu.cn/docs/20200227111544292237.pdf (accessed on 4 May 2021)). The testes and epididymis were dissected out. The left testis and epididymis were fixed in 10% formaldehyde for histological processing, while the right testis and epididymis were stored at −80 °C for evaluation of daily sperm production (DSP) and comet assay.

### 2.4. The Relative Weight of Testes

During the animal trials, the weight of animals was checked after every 4 days to update the Cd doses. After dissection of animals, the relative weight of testis was calculated by dividing testis weight (mg) by animal weight (g).
Relative weight of testis (mg/g)) = testis weight (mg)/animal weight (g)
whereas the total weight gain/loss was calculated by subtracting the final weight from the initial weight.
Weight gain/loss (g) = final weight (g) − initial weight (g)

### 2.5. Histology

The testes and epididymis were fixed in 10% formaldehyde and embedded in paraffin wax for slides preparation. In brief, 5 µm-thick sections were cut from the paraffin blocks using a microtome. These sections were affixed on glass sides, on a slide warmer, and deparaffinized prior to staining with hematoxylin and eosin (H&E) stain. The slides were examined under an Olympus microscope (Model IX2-ILL100) equipped with a micro-photographic system.

### 2.6. Image J Software Application

During histological analysis of the testis, five parameters (i.e., interstitial space, thickness of tunica albuginea, diameter of the seminiferous tubules, diameter of the tubular lumen, and height of the epithelium) were measured. In the histological analysis of the epididymis, the area and diameter of the lumen and the height of the epithelium were measured. All the measurements were obtained using Image J software. The standard scale (Appendix A) used for image J was taken with the same magnification as the remaining pictures.

### 2.7. Daily Sperm Production

Testicular tissues stored at −80 °C were defrosted at room temperature for 2–5 min prior to homogenization. Spermatids that were resilient to homogenization were calculated according to the protocol of Robb et al. [35]. The testes were weighed, and tunica albuginea was removed. Approximately 100 mg of testis parenchyma was homogenized in 2 mL of saline and diluted, according to Jahan et al. [36]. A small portion (5.5 µL) of the sample was placed in Neubauer chambers (hemocytometer), and late spermatids were counted under a microscope at 40× magnification. DSP was calculated according to the following formula.
Y=(x/16)×100×5×5.5×1000
where *Y* is the total number of spermatids, x is the number of spermatids counted on hemocytometer, 16 is the total number of squares observed, 100 is the total number of squares, 5 is the dilution factor, 5.5 µL is the sample volume loaded into the hemocytometer, and 1000 is the conversion factor from microliters to milliliters.

### 2.8. Comet Assay

Damage to heritable DNA was determined using comet assay [37], with some modifications. Slides were prepared by placing 100 µL of 1% regular melting point agarose and covered by a large coverslip. The slides were then placed in a refrigerator for 30 min to solidify the agarose. After 30 min, the slides were placed at a slide warmer at 37 °C and the coverslips were carefully removed. Next, 20 µL of a suspension of sperm from the cauda epididymis and 65 µL of low-melting-point agarose were mixed in Eppendorf using a micropipette. The mixture was placed on the agarose slides, and a coverslip was used to spread it. The slides were placed in a wooden slide box to avoid exposure to direct light and the resultant DNA damage, and the agarose slides were solidified in the refrigerator. After solidification, the coverslips were removed and the slides were submerged in staining jars containing freshly prepared cold lysing solution (100 mM EDTA disodium salt, 10 mM Tris, 2.5 M NaCl, pH 10, with 1% Triton X-100 added just before use). The slides were soaked in this solution overnight, and the staining jars were covered with aluminum foil to avoid direct exposure to light and DNA damage. A gel electrophoresis tank (horizontal) was filled with electrophoresis solution (300 mM NaOH + 1 mM EDTA, pH 12.5), and 12 slides were placed in it side by side in two rows, with the agarose end facing the positive terminal. The slides were left in the tank for some time, and the DNA fragments were separated by electrophoresis for 10 min at 25 V and 300 mA. The alkaline detergent was washed after electrophoresis with 0.4 M Tris solution to avoid interactions with the stain.

Exactly 100–200 mL of 20 mg/mL acridine orange solution was overlaid on the slides by using a coverslip for comet scoring. The slides were observed under a fluorescent microscope, and comets were analyzed using Casplab_1.2.3b2.

### 2.9. Statistical Analysis

One-way analysis of variance followed by Tukey’s test was applied to compare the experimental data of different groups by using Graphpad Prism 5 software. All results are presented as mean ± SEM, and the significance level was set to * *p* < 0.05, ** *p* < 0.01.

## 3. Results

In the current experiment, significant weight loss was observed in Cd-treated groups (Figure 1). The weight of animals in the control group was increased by 28 g, while the weight in the 15 mg Cd/kg treatment group decreased by the same amount (Table 1, Figure 1). The total weight gain in the 5 mg Cd/kg treatment group was similar to that in the control group. The weight of the testes and epididymis showed no variations among the control and Cd-treated groups (Table 1). However, the relative mass of the testes was significantly higher in the 15 mg Cd/kg treatment group compared to that in the control group (*p* < 0.05) and mg Cd/kg treatment groups (*p* < 0.01) (Table 1).

### 3.1. Histology

#### 3.1.1. Testes

Microscopic analysis of the testes showed that the control group had closely arranged seminiferous tubules with normal spermatogenesis (Figure 2A,B). Germ cells of all stages were observed in germinal epithelium, and the lumens of the tubules were narrow and filled with sperm. In brief, all the tubules observed in the control group could be divided into two categories based on the morphologic appearance, the first showing the early (initial) half of the spermatogenic cycle (stage 1–8) and the 2nd showing the other half (stage 9–14). In interstitial space, Leydig cells of different shapes (round, oval, and irregular) were present, along with blood vessels surrounded by seminiferous tubules (Figure 2A,B). The diameter of the seminiferous tubules in the Cd-treated groups showed variations from the control group (Figure 2C,D, Table 2), indicating decreased spermatogenesis. The high-dose groups (10 and 15 mg Cd/kg/day) showed the greatest deterioration in testicular tissues. The size of the interstitial space (Table 2), the height of the epithelium (Figure 2), and the thickness of the tunica albuginea (Figure 2 and Table 2) showed remarkable variations compared to the control group (Table 2).

The thickness of the tunica albuginea (Figure 2) in the 10 mg Cd/kg treatment group decreased significantly (*p* < 0.001) compared with that in the control group. The effect of Cd on the tunica albuginea was dose-dependent, as evidenced by the lack of a significant difference between the control and 5 mg Cd/kg treatment groups. The exterior walls of the tunica albuginea in animals in the 15 mg Cd/kg treatment group were affected in a non-continuous manner, where the wall was thicker at some points and thinner at others (Figure 2H). Overall, the mean thickness was similar to the control. However, as can be seen in Figure 2H, the tunica albuginea was severely affected, and the size of the wall was significantly (*p* < 0.001) thicker than that in the 10 mg Cd/kg treatment group (Table 2, Figure 2).

The average space between different seminiferous tubules (Figure 2) showed no significant (*p* > 0.05) difference between the 5 mg Cd/kg treatment and control groups. The interstitial space significantly decreased in the 10 and 15 mg Cd/kg treatment groups (*p* < 0.001) compared with that in the control and 5 mg Cd/kg treatment groups (Figure 2E,G). The number of Leydig cells in the interstitium was similar in all groups, but the interstitial space in Cd-treated groups was remarkably reduced compared with that in the control group (Figure 2, Table 2). The average diameter of seminiferous tubules increased with increasing Cd concentration. In the 10 mg Cd/kg treatment group, the diameter of tubules increased significantly (*p* < 0.001) compared with that in the control and 5 mg Cd/kg treatment groups. All other groups showed increases in the diameter of seminiferous tubules, but the differences noted were not significant (*p* < 0.05).

The epithelial height and tubular lumen showed interesting results. The average height of the epithelium (both early and late phase of spermatogenesis) increased significantly (*p* < 0.001) in the 10 and 15 mg Cd/kg treatment groups compared with that in the control group (Figure 2). A significant increase in epithelium height (*p* < 0.05) was also noted compared with that in the 5 mg Cd/kg treatment group (Table 2), but the process of spermatogenesis appeared to be impaired in all Cd-treated groups (Figure 2). It was observed in the current study that the epithelium of the control group was much denser, having cells of all stages of spermatogenesis (Figure 2A), whereas, in Cd treated groups, the overall number of cells appear to be much lower compared to the control group (Figure 2C,E,G). The DSP showed non-significant variation (Table 1), which could be explained by the histological deformities observed here. The size of the tubular lumen diameter in the 15 mg Cd/kg treatment group showed a significant difference compared with that in the control and the two other Cd-treated groups. The spermatozoa in the lumen of all Cd-treated groups were premature, thus demonstrating the marked impact of Cd on spermatogenesis.

#### 3.1.2. Epididymis

Histological analysis of the epididymis showed that the 2D area of the tubular lumen significantly increased with increasing Cd dose. In the 5 mg Cd/kg treatment group, the 2D area of the lumen slightly increased, but this increase was not significant. However, a significant increase in the area of the tubular lumen of the 10 mg Cd/kg and 15 mg Cd/kg treatment groups in comparison with the control and 5 mg Cd/kg treatment groups was observed. The diameter of the lumen significantly increased in all Cd-treated groups, but the increase in the 10 mg Cd/kg treatment group was the highest. The epithelium showed no remarkable difference among all groups (Figure 3).

### 3.2. Daily Sperm Production

The mean value of DSP in the Cd-treated groups slightly decreased compared with that in the control group (Table 1). Among the groups assessed, the 10 mg Cd/kg treatment group revealed the lowest amount of sperm produced. However, overall, no significant difference in the amount of sperm produced was found among the groups.

### 3.3. Comet Assay

Damage to heritable DNA was determined by comet assay. The results of the Cd-treated and control groups are presented in Table 3, and relevant microphotographs are shown in Figure 4. The comets produced in different groups varied. The numbers of comets observed in each group are not shown in this paper because some of the comets may have been washed away by prolonged soaking in the buffer. Overall, the number of comets observed in the control group was lower compared with that found in the Cd-treated groups. DNA damage was estimated by considering different parameters (i.e., comet length, head length, tail length, % DNA in head, % DNA in tail, and tail moment).

The head and comet lengths showed significant increases. The length of the head in the 5 mg Cd/kg treatment group was significantly (*p* < 0.001) shorter than that in the control group, whereas, in 10 mg/kg, it was *p* < 0.05 and 15 mg/kg *p* < 0.01 (Table 3). The percentage of DNA in the tail was lower in the control group than in the treatment groups, but the difference observed was not significant (*p* < 0.05). The lengths of the comet and head were comparable among all Cd-treated groups (*p* < 0.01 for the 5 mg Cd/kg treatment group and *p* < 0.05 for the 10 and 15 mg Cd/kg treatment groups when compared with the control) (Table 3, Figure 4).

## 4. Discussion

A major concern related to the increase in the global population is heavy metal toxicity. Cd exposure occurs through water, food, and air. In previous literature, the average Cd absorbed from food in some parts of the world was approximately 15.5 µg/day, and the average Cd contents in the blood and urine are 0.74 µg/L and 0.34 µg/g, respectively [1,38,39]. The results of previous molecular cell biological experiments indicate that Cd has more than one complex effect on different cells and pathways, especially the pituitary-hypothalamus sex organ pathway [1,40,41]. Cd disturbs cell proliferation, differentiation, cell cycle progression, and DNA replication and repair; the apoptotic pathways were also impaired [1,42,43,44]. In the present study, we observed the microscopic damage caused by oral administration of Cd on the reproductive system and the process of spermatogenesis in male SD rats.

A 28 g increase in the weight of the control group was noted (Figure 1, Table 2). Rats in the high-dose Cd treatment groups (10 and 15 mg Cd/kg) lost approximately 30 g of weight (Table 2, Figure 1). Rats in the low-dose Cd treatment group (5 mg Cd/kg) showed a weight gain similar to that in the control group. Hence, besides initiating reproductive toxicity, Cd affects the weight of animals. This finding contradicts some of the data reported in the literature [44,45]. According to Sagba et al. [46], Cd has a negative effect on weight gain. In our findings, the effect of Cd on weight gain was not significant, and these findings are in accordance with references having weight loss, for example. Leach et al. and Santose et al. [46,47,48]. However, when we compared the difference in weight gain, the result became interesting due to a significant variation in the control and Cd-treated groups.

In the current study, major Cd-induced damage was investigated in the reproductive system of rats. No variations in the weight of the testes or epididymis were noted. Earlier studies reported contrasting results on the weight of reproductive organs in animals exposed to Cd [48,49,50]. According to some research groups, Cd ions exert marked effects on the weight of reproductive organs, as well as the bodyweight of the animals; specifically, the weight of the reproductive organs significantly decreased following Cd exposure. Some research groups reported no change in accessory glands [51,52], and our findings appear to agree with these groups, i.e., the weight of the animals decreased but no significant difference in the weight of the testes and epididymis was noted. One of the major factors in weight gain could be the route of administration. In experimental studies, the path of administration affects the delivery and absorption of Cd [9,24]. According to Ryan, Wilhelm, Ysart, and their respective teams, the absorption of Cd is much higher when the route of administration is i.p. injection, compared to intestinal absorption or occupation and cigarette smoke inhalation [23,25,26]. In the current experiment, the animals were exposed to Cd through oral gavage, and Cd absorbed by the intestine had minimal effect on the weight of accessory organs.

Disparities in morphology are directly related to physiological problems [53] and decreased DSP; these disparities result in issues related to spermatogenesis, which is fulfilled in the epithelium of seminiferous tubules [54,55,56]. We performed histopathology of the testes to determine morphological damage in these organs. We observed several deformities in the testicular seminiferous tubules of Cd-treated groups, consistent with the findings of previous research groups [57,58,59,60]. According to one research group [61], pro-inflammatory cytokines stimulate the inflammatory process, such as vascular congestion, interstitial mononuclear cell infiltrations, tissue degeneration, and necrosis. Figure 2 reveals the presence of blood cells in a cluster, as well as damage to the epithelium, thus indicating the onset of necrosis. Increases in the diameter of the seminiferous tubules and length of the epithelium (Table 2) were also observed, thus indicating the beginning of inflammation. Despite the variations in histopathology noted in this work, the daily sperm produced was not in accordance with that reported in previous studies. In the current study, we observed a non-significant decrease in DSP. However, morphological changes are directly related to physiological changes [53]. Cd-induced oxidative stress has been reported to be the chief source of decreases in sperm count [62]. Some researchers speculate that genome instability and DNA damage may sometimes result from Cd toxicity, resulting in malignancy [10,40,63,64]. Correction of inaccurate base pairs, deletion of unrequired base pairs, and repair of damaged pairs are usually blocked by Cd via the recognition of damaged DNA and attachment of proteins to affected sites [10,40,65,66]. Our results are in accordance with those studies. In the present study, the comet and head length were remarkably extended, and a higher percentage of DNA in the tail of comets was found in the Cd-treated groups, which points out the breaks in heritable DNA.

## 5. Conclusions

Observation of various parameters led us to conclude that higher concentrations of Cd in food and water could result in reproductive deformities, along with other damages to the reproductive and body physiology.

## Figures and Tables

**Figure 1 ijerph-18-06038-f001:**
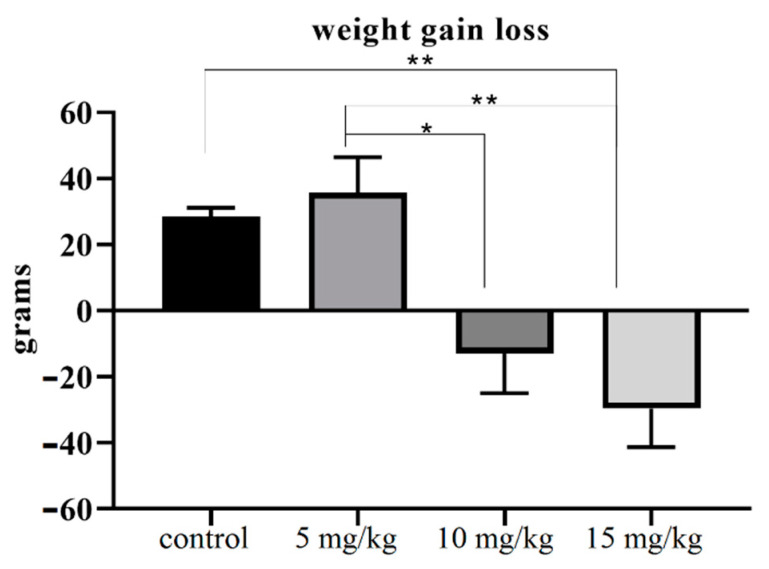
Mean (with SEM) weight gain/loss during animal trials, showing about 30 g decrease in weight of 15 mg/kg animals, whereas the same weight was gained in the control group. (Probability: * = *p* < 0.05 and ** = *p* < 0.01)

**Figure 2 ijerph-18-06038-f002:**
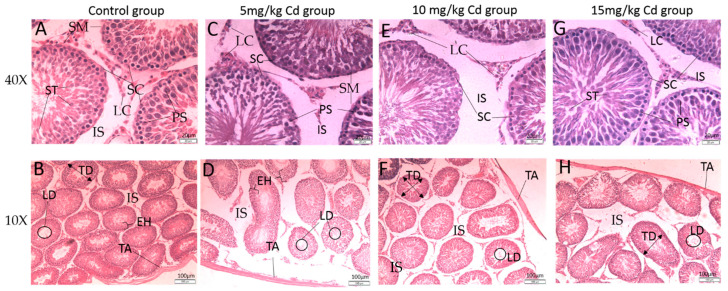
Microphotograph of testicular tissue (seminiferous tubules). (**A**,**B**) The control group showed normal seminiferous tubules with a lumen filled with sperm. The interstitial space revealed normal Leydig cells. (**C**,**D**) The 5 mg Cd/kg treatment group showed a decrease in epithelial height and a marked increase in interstitial space. The (**E**,**F**) 10 mg Cd/kg and (**G**,**H**) 15 mg Cd/kg treatment groups revealed prominent damage. In particular, the number of Leydig cells in interstitial space are decreased and spermatogenesis in epithelium appeared to be disrupted. (LC: Leydig cells, SM; smooth muscle, ST: spermatids, PS: primary spermatocytes, IS interstitial space, SC: Sertoli cells, LD: (lumen diameter) tubular lumen, EH: epithelial height, TD: diameter of seminiferous tubules TA: tunica albuginea.

**Figure 3 ijerph-18-06038-f003:**
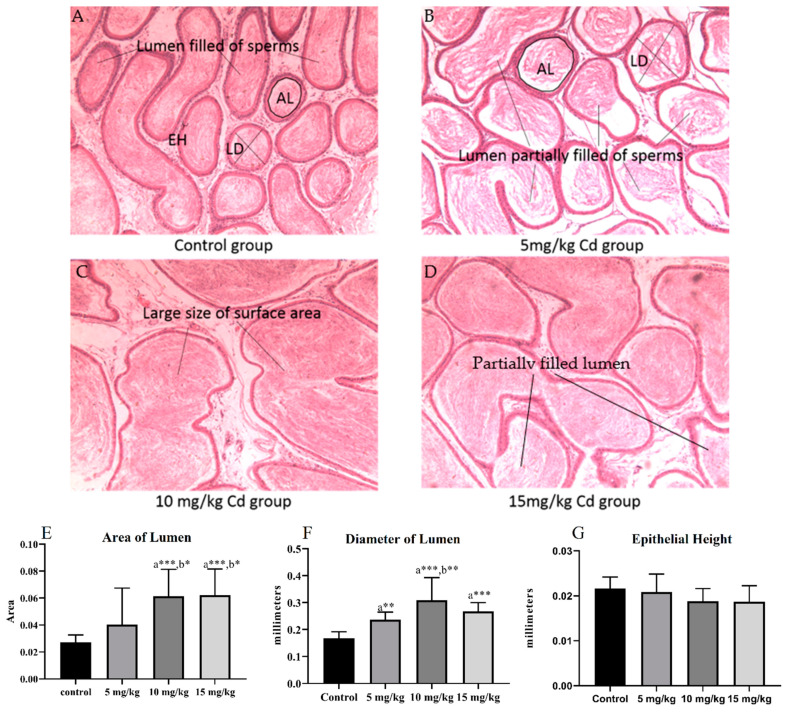
Microphotographs of the epididymis. (**A**) The control group showed normal morphology and lumen full of sperm. (**B**) The 5 mg Cd/kg treatment group showed damaged tubules with a partially empty lumen. (**C**) The 10 mg Cd/kg and (**D**) 15 mg Cd/kg treatment groups showed lumen with a large surface area. The (**E**) area and (**F**) diameter of the lumen. (**G**) Height of the epithelium (magnification 10×). (a = comparison to control, b = comparison to 5 mg/kg group. Probability: * = *p* < 0.05, ** = *p* < 0.01 and *** = *p* < 0.001).

**Figure 4 ijerph-18-06038-f004:**
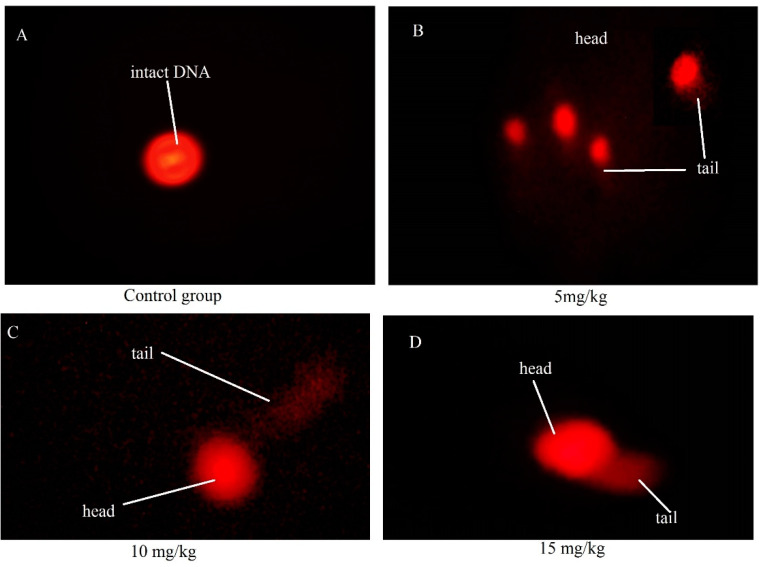
Fluorescent microphotograph of rat sperm after Cd treatment as determined by single-cell gel electrophoresis with acridine orange staining. (**A**) Control group presenting intact DNA, (**B**) 5 mg Cd/kg treatment, (**C**) 10 mg Cd/kg treatment, and (**D**) 15 mg Cd/kg treatment groups showing damaged DNA represented by the tail.

**Table 1 ijerph-18-06038-t001:** Mean and SEM of body weights, testes weight and epididymis weights. Daily sperm production per 100 mg of the testis, and the relative weight of testis.

Groups (*n* = 4)		Control (6 Animals)	5 mg/kg (6 Animals)	10 mg/kg (6 Animals)	15 mg/kg (6 Animals)
DSP (×10^6^/100 mg)		14.53 ± 1.17	12.07 ± 12.84	11.51 ± 0.17	13.05 ± 0.7
Body weight (grams)	Initial	238.25 ± 6.46	254 ± 4.97	252.5 ± 14.26	287.5 ± 14.26
	Final	266.75 ± 6.69	289.75 ± 6.41	235.5 ± 9.5	253.67 ± 6.89
Weight of testis (grams)	Right	1.47 ± 0.05	1.92 ± 0.075	1.85 ± 0.05	1.73 ± 0.03
	Left	1.47 ± 0.05	1.9 ± 1.22	1.85 ± 0.15	1.76 ± 0.03
Relative weight of testis (mg/g)		1.75 ± 0.02	1.85 ± 0.19	1.68 ± 0.12	2.06 ± 0.02 ^a^*^,c^**
Weight of epididymis (g)	Right	0.63 ± 0.09	0.97 ± 0.02	0.71 ± 0.01	0.83 ± 0.32
	Left	0..63 ± 0.09	0.89 ± 0.03	0.69 ± 0.011	0.70 ± 0.04

All values are expressed as mean ± SEM (^a^ = comparison to control, ^c^ = comparison to 10 mg/kg group. Probability: * = *p* < 0.05 and ** = *p* < 0.01).

**Table 2 ijerph-18-06038-t002:** Mean ± SEM of different parameters of testicular histology.

Groups	Interstitial Space	Thickness of the Tunica Albuginea (µm)	Diameter of the Seminiferous Tubules (µm)	Height of the Epithelium (µm)	Diameter of the Tubular Lumen (µm)
Control	63.58 ± 4.61	36.45 ± 1.5	235.34 ± 5.95	70.98 ± 2.02	92.51 ± 3.50
5 mg/kg	60.45 ± 4.54	33.33 ± 1.53	244.04 ± 4.73	76.13 ± 2.71	95.30 ± 5.81
10 mg/kg	23.96 ± 2.67 ^a^***^, b^***	23.26 ± 0.67 ^a^***^, b^***	273.6 ± 7.17 ^a^***^, b^***	89.72 ± 3.53 ^a^***^, b^**	90.59 ± 3.33
15 mg/kg	24.58 ± 2.84 ^a^***^, b^***	36.23 ± 3.2 ^c^***	258.14 ± 7.67	92.58 ± 3.08 ^a^***^, b^**	53.02 ± 2.44 ^a^***^, b^***^,c^***

All values are expressed as mean ± SEM (^a^ = comparison to control, ^b^ = comparison to 5 mg/kg group, ^c^ = comparison to 10 mg/kg group. Probability: ** = *p* < 0.01 and *** = *p* < 0.001)

**Table 3 ijerph-18-06038-t003:** Mean ± SEM of DNA damage in control and Cd treated adult rats after 17 days of treatment.

Parameter	Control	5 mg/kg	10 mg/kg	15 mg/kg
Head Length (µm)	231.86 ± 34.47	122.23 ± 9.38 ***	156.93± 15.73 *	131.0 ± 11.97 **
Tail Length (µm)	7.29 ± 1.70	11.08 ± 1.55	9.8 ± 1.96	9.59 ± 1.15
Comet Length (µm)	239.14 ± 34.32	133.30 ± 8.85 **	166.8 ± 15.11 *	166.06 ± 13.57 *
% DNA in Head	97.93 ± 0.95	96.58 ± 0.62	96.58 ± 0.62	95.23 ± 0.62
% DNA in Tail	2.07 ± 0.95	3.42 ± 0.62	3.32 ± 0.90	4.76 ± 0.62
Tail Moment	0.39 ± 0.15	0.48 ± 0.15	0.53 ± 0.25	0.53 ± 0.12

(Probability: * = *p* < 0.05, ** = *p* < 0.01, *** = *p* < 0.001, * = significant difference compared to the control).

## Data Availability

No data set is linked to current study; the summary of experiments performed is presented in current manuscript.

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
