# Peer review of "Damage to the Testicular Structure of Rats by Acute Oral Exposure of Cadmium"

_ijerph, 2021, doi:10.3390/ijerph18116038_

Round 1
Reviewer 1 Report
Manuscript Reference: IJERPH-1158375
Title: Damage to the Testicular Structure of Rats by Acute Oral Exposure of Cadmium
This manuscript presents an interesting study about the effect of oral exposure to Cd on spermatogenesis. Therefore, it falls within the scope of this journal. Therefore, I recommend the publication of the manuscript with minor review.
Line 61-62: Please, delete “In this work, Cd was administered orally using cadmium chloride (CdCl2) solution to adult SD rats for 17 days (more than acute exposure). And the effects of Cd on testis and sperm maturation and damage to heritable DNA were studied.”, because this part has to be in M&M
Line 92: In agreement with line 89, authors would have to change “for 17 days (acute exposure)” to “for 17 days (intermediate exposure)”
Figure 1 shows same information that table 1. Please, authors have to eliminate the first row of table 1 or figure 1.
Author Response
Line 61-62: Please, delete “In this work, Cd was administered orally using cadmium chloride (CdCl2) solution to adult SD rats for 17 days (more than acute exposure). And the effects of Cd on testis and sperm maturation and damage to heritable DNA were studied.”, because this part has to be in M&M
Response: Thank you for your suggestion, these sentences have been deleted in revised manuscript
Line 92: In agreement with line 89, authors would have to change “for 17 days (acute exposure)” to “for 17 days (intermediate exposure)”
Response: The phrase has been changed to (intermediate exposure)
Figure 1 shows same information that table 1. Please, authors have to eliminate the first row of table 1 or figure 1.
Response: Thanks for pointing out, line one of the table 1 has been removed in revised manuscript

Reviewer 2 Report
The study by Tariq Iqbal and collaborators describes the adverse effect of Cd on rat testicular structure after an acute exposure.
The study is interesting, clearly written, results support conclusions.
Maybe it would deserve a minor revision as to English language.
Author Response
Maybe it would deserve a minor revision as to English language.
Response: Thanks for the suggestion, the manuscript is revised thoroughly
Reviewer 3 Report
The manuscript was poorly written and full of errors.
- EPA’s oral reference dose in water is 5E-4 mg/kg/day (using proteinuria as the endpoint). So the doses used in this manuscript were much higher than EPA’s reference dose, so the exposure is not realistic.
- Line 68: “one group” per cage?
- Line 93: “acute exposure”. 17-days exposure is not ‘acute exposure’
- Line 98: “5mm-thick”?
- Line 104: Why “Turkey’s test” was used as post hoc test?
- The authors need only to keep Table 1 or figure 1. They contain the same information.
- Table 2: impact on tunica albuginea. It looks like the effect was U shaped not linear. Authors need to discuss this.
- Table 2 and Figure i-ix contain the same information. Only one needs to be retained.
- Lines 197-198: was exterior wall of 15 mg/kg group thicker than 10 mg/kg?
- Lines 201-202; was the interstitial space in 15 and 10 mg/kg group increased compared to control and 5 mg/kg group
Author Response
Reviewer 3
- EPA’s oral reference dose in water is 5E-4 mg/kg/day (using proteinuria as the endpoint). So the doses used in this manuscript were much higher than EPA’s reference dose, so the exposure is not realistic.
Response: Thank you for your comment, in current experiment the dose selection was based on ATSDR (Agency of Toxic Substances and Disease Registry)
- Line 68: “one group” per cage?
Response: the description is removed in revised manuscript
- Line 93: “acute exposure”. 17-days exposure is not ‘acute exposure’
Response: Thank you. The description has been changed to “intermediate exposure” according to ATSDR
- Line 98: “5mm-thick”?
Response: Sorry for typing error, this is 5 micrometres (µm)
- Line 104: Why “Turkey’s test” was used as post hoc test?
Response: We used Tukey’s post hoc test, as it compares the mean of every treatment group to the mean of all the remaining groups. All the comparison are pairwise with the identification of any difference between 2 means which is greater than the expected standard error.
- The authors need only to keep Table 1 or figure 1. They contain the same information.
Response: Thank you! Table 1 has been edited
- Table 2: impact on tunica albuginea. It looks like the effect was U shaped not linear. Authors need to discuss this.
Response: Thank you for your suggestion. The mean results of the 15mg/kg group are similar to the control, but this is because, the damage in 15mg/kg was severe, at some points the wall size was increased and at some point it was either same to control or thinner than it. But the overall mean was somewhat similar to control group. more explanation has been added as per your instruction. Also figure 3 was added in first draft because of results like this.in revised manuscript figure 3 is merged in figure 2 to easily understand this point
- Table 2 and Figure i-ix contain the same information. Only one needs to be retained.
Response: Thank you! figure 2 has been edited. And part related to table 2 is deleted
- Lines 197-198: was exterior wall of 15 mg/kg group thicker than 10 mg/kg?
Response: (answer is in question 7)
- Lines 201-202; was the interstitial space in 15 and 10 mg/kg group increased compared to control and 5 mg/kg group
Response: Response to Question 9 and 10
Figure 3 is added because of these reasons. The rest of the parameters could be explained by figure 2.

Reviewer 4 Report
The paper is not full of innovation, the experimental design and the description of this article also needs to be improved more convincingly.
1.The relationship between cadmium and testicular structure has already been discussed by some researchers, thus innovation is not enough,even if acute exposure is the starting point and is not persuasive.
2. the paper is poorly written.
1) Line 9“the worst effect” the definition of “the worst effect”is unclear.
2) Line 61-62“for 17 days (more than acute exposure). ” it is somehow unproper in the introduction part.
3) Line 68“one group”per cage? at “a” controlled temperature ?
4) Figure 1 shows the same information as table 1.
5) Line 91-92“for 17 days (acute exposure)”17-day exposure is not acute exposure.
6) Line 265-267 the description needs to be more concise.
7) Line 303-304 “if the concentration of Cd in food and water is higher, it results in reproductive deformities”the time factor is not taken into account as "acute exposure" is related to time. Thus the conclusion part needs to be improved.
Author Response
Reviewer 4
- The relationship between cadmium and testicular structure has already been discussed by some researchers, thus innovation is not enough,even if acute exposure is the starting point and is not persuasive.
Response: Different research group have different approach in studying a similar problem. The current study was designed according to guidelines of ATSDR (Agency of Toxic Substances and Disease Registry). Speaking of oral exposure, no research group mentioned the DNA damage in heritable DNA when the exposure is oral. In most cases the damage in DNA is discussed, whereas in this study the damage in heritable DNA is discussed. The daily sperm production is a very old parameter, which is seldomly used. Usually the old parameters are ignored completely by the research community. We take the protocol of Rob et al.,1970. And used that to study the effect of Cd on sperm production. Which is studied for very few chemicals.
- the paper is poorly written.
Response: The revised manuscript has been edited by a professional proofreader
- Line 9“the worst effect” the definition of “the worst effect”is unclear.
How about “the most severe effect”
- Line 61-62“for 17 days (more than acute exposure). ” it is somehow unproper in the introduction part.
Response: In the revised manuscript, the description has been changed to “intermediate exposure” according to ATSDR standard.
- Line 68“one group”per cage? at “a” controlled temperature ?
Response: The description has been removed in revised manuscript.
- Figure 1 shows the same information as table 1.
Response: Table 1 is edited and mentioned line is removed
- Line 91-92“for 17 days (acute exposure)”17-day exposure is not acute exposure.
Response: Thank you. The description has been changed to “intermediate exposure” as per ATSDR
- Line 265-267 the description needs to be more concise.
Response: Thank you. The description has been revised and more references have been added
- Line 303-304 “if the concentration of Cd in food and water is higher, it results in reproductive deformities” the time factor is not taken into account as "acute exposure" is related to time. Thus the conclusion part needs to be improved.
Response: Conclusion is changed

Reviewer 5 Report
In their paper, Tariq Iqbal and colleagues describe the testicular damage found in the testis and epididymis of Sprague–Dawley rats treated with three different doses of cadmium chloride for 17 days, by comparing the histological features in control and treated rats. The MS might be of interest to those working in the field of environmental toxicants. However, the M&M section needs to be properly detailed, so the reader may understand what kind of essays/parameters were tested and how. This is particularly important for the histological studies and the (epididymal ?) sperm evaluation. The parameters seem somehow different when mentioned in M&M and in Results. Also, the information regarding the statistical analysis must include additional information (do data follow the assumption of a normal distribution?)
When describing the impact of putative disruptors in the testicular function, authors should use a very strict approach. Spermatogenesis should be properly evaluated (which would imply assessing the effects on the spermatogenic cycle) and the appearance of Sertoli and Leydig cells should be detailed. Please address some additional references in the topic:
https://journals.sagepub.com/doi/pdf/10.1080/01926230290105695
https://onlinelibrary.wiley.com/doi/abs/10.1002/bdrb.10041
I have some concerns regarding the description of the histologic changes, as in the commented file I attach to this revision.
Also, and coping with the changes to be introduced in the Results section, the discussion should be strengthened. A few recently published papers are available on the reproductive toxicity of cadmium that could be used herein.
The histological images are not adequate. If Hematoxylin-Eosin staining was used (as stated in M&M), then the tissues should present a different color and the cell nuclei should appear contrasted. Furthermore, the magnification does not allow to see the "deformities" (??) of the epithelium, as described in the figure caption. Furthermore, the deformities should be contextualized. I am not sure that I saw the deformities the authors planed to be seen.
Some images can be removed as they duplicate the information provided in tables or in other images.
Finally, the reference list needs the authors' attention. several references fail to provide the journal identification
Best regards

Author Response
Reviewer 5
In their paper, Tariq Iqbal and colleagues describe the testicular damage found in the testis and epididymis of Sprague–Dawley rats treated with three different doses of cadmium chloride for 17 days, by comparing the histological features in control and treated rats. The MS might be of interest to those working in the field of environmental toxicants. However, the M&M section needs to be properly detailed, so the reader may understand what kind of essays/parameters were tested and how. This is particularly important for the histological studies and the (epididymal ?) sperm evaluation. The parameters seem somehow different when mentioned in M&M and in Results. Also, the information regarding the statistical analysis must include additional information (do data follow the assumption of a normal distribution?)
Response: in material and method section each and every point in described under separate heading (for example the detail of animals and its environment, the experimental design and the duration of experiment) the reasons for current design and duration of experiment are explained in brief detail under mentioned headings. Later on the protocols of performed parameters are given in detail. The parameters in histology are explained in “Application of Image J software”
When describing the impact of putative disruptors in the testicular function, authors should use a very strict approach. Spermatogenesis should be properly evaluated (which would imply assessing the effects on the spermatogenic cycle) and the appearance of Sertoli and Leydig cells should be detailed. Please address some additional references in the topic:
when we study spermatogenic cycle, we usually follow OECD guidelines, the current experiment is designed according to ATSDR. And the trails were for duration of 17 days. Because of which we cannot mention spermatogenic cycle. And the focus is acute damages.
https://journals.sagepub.com/doi/pdf/10.1080/01926230290105695
https://onlinelibrary.wiley.com/doi/abs/10.1002/bdrb.10041
Thanks for suggestion .
I have some concerns regarding the description of the histologic changes, as in the commented file I attach to this revision.
(points about histology in commented file )
- for sure, this decritpion does not apply to all the tubules, which would present different stages of the spermatogenic cycle. How did the authors account for this fact?
Response : in the mentioned sentences, the roughly round tubules are discussed. In histology section the calculations of all parameters (except exterior wall and interstitial space) are done in roughly round tubules. And elongated tubules are not taken into account.
- If disruption of psermatogenesis has been observed, in order to report it adequatly, authors should thoruoughly describe it, including by mention the changes in the spermatogenic cycle that existed.
Comparisons to the normal stages might be needed.
Response: to study the complete spermatogenic process, the duration of animals trails should be almost 2 months (OECD guidelines). we studied the acute damages, as per ATSDR division of Cd exposure.
- the figures need to be replaced. some are unfocused or fuzzy.
Response: the images in figures 2 and 3 are merged into single figure. And some images are replaced
- but in the lesgend something was said aboit existing edema in some groups? which parameter in the interstitium are we talking in here?
Response: here the interstitial space/length between seminiferous tubules is discussed
- usually, when spermatogenesis is impaired, the height of the spermatogenic epithelium is reduced. better characterization of the changes are needed in here
Response: the stain absorbed by tubular lumen is different in control and Cd treated groups. where the lumen of Cd treated groups absorbed lesser stain pointing out that sperm content (number) is much lesser as compared to control group.
In future study the epethilial height, sperm count as well as the molecular parameters will be studied to understand the effect on these various parts
The methodology portion is corrected
Also, and coping with the changes to be introduced in the Results section, the discussion should be strengthened. A few recently published papers are available on the reproductive toxicity of cadmium that could be used herein.
Response: thanks for suggestion
The histological images are not adequate. If Hematoxylin-Eosin staining was used (as stated in M&M), then the tissues should present a different color and the cell nuclei should appear contrasted. Furthermore, the magnification does not allow to see the "deformities" (??) of the epithelium, as described in the figure caption. Furthermore, the deformities should be contextualized. I am not sure that I saw the deformities the authors planed to be seen.
Response: in current experiment the whole histology was performed in the lab. Hematoxylin stain used was of “Beijing Dingguo Changsheng Biotechnology Co. LTD” (http://www.dingguo.com/) and the eosin used was of “Sinopharm Chemical Reagent Co.,Ltd”( https://www.reagent.com.cn/). As the histology s performed by us,
Some images can be removed as they duplicate the information provided in tables or in other images.
Finally, the reference list needs the authors' attention. several references fail to provide the journal identification
Response: all the references were given using mendely Software. The mentioned refrences are updated from google scholar.
Response: in material and method section each and every point in described under sepate heading (for example the detail of animals and its environment, the experimental design and the duration of experiment) the reason for current design and duration of experiment is explained under mentioned headings.
Point 2: when we study spermatogenic cycle, we usually follow OECD guidelines, the current experiment is designed according to ATSDR. And the trails were for duration of 17 days. Because of which we cannot mention spermatogenic cycle
Point 3: some new refrences are added to discussion section
Point 4: the histology was performed according to standard protocol. Where the tissue was dehydrated with different grades of ethanol, and affixed in wax. After the sectioning the slides were stained with H and E.

Reviewer 6 Report
In this manuscript, the authors evaluated the reproductive toxicity of Cadmium in Male Sprague-Dawley rats. To achieve their aims the authors used three doses of cadmium and performed its administration by gavage. In my opinion, this manuscript is interesting but not innovative. So, it needs do be improved:
- The abstract should include the cadmium doses and numerical results
- How were these animals sacrificed?
- What food was administrated to these animals? What is the food reference? What is the food composition?
- Cadmium was administrated during 17 consecutive days? Including the weekends?
- How were these animals sacrificed?
- During the experimental assay what biological variables were registered? Food consumption? Water consumption? Body weight variation?
- In my opinion, points 2.6 and 3.2 are not Daily Sperm Production, but Sperm Production at the end of the experimental protocol, ie, at animals' sacrifice.
- Figure 1 repeats the information placed in table 1. So, figure 1 should be removed.
- Authors should place on table 1 the relative weight of testis. The relative weight of testis is calculated by dividing testis weight/animal weight.
- How was calculated: Weight gain/loss?
- What is the impact of three different doses of cadmium on animals' food and water consumption?
- Were collected blood samples? And Kidneys? What are the impacts of cadmium on kidney function: kidney histology and creatinine?
- What is the real impact of this trial for future work? What is its innovation?
Author Response
Reviewer 6
- The abstract should include the cadmium doses and numerical results
Response: The numerical doses are added, and summary of result is already given
- How were these animals sacrificed?s
Response: The animals were anesthetized using 750mg/kg of intraperitoneal injection of 2,2,2-tribromoethanol solution before they were sacrificed.
- What food was administrated to these animals? What is the food reference? What is the food composition?
Response: The standard food was used through out the experiment. The detailed composition of food is mentioned in table blow
|
Beijing Keaohui Feed Co., Ltd |
||||||
|
Product component analysis guaranteed value(Content per kilogram of feed) |
||||||
|
Vit. A |
≥14000IU |
Na |
≥2g |
crude protein |
≥200g |
|
|
Vit. D |
≥1500IU |
K |
≥5g |
crude fat |
≥40g |
|
|
Vit. E |
≥120IU |
Mg |
≥2g |
crude fibre |
≤50g |
|
|
Vit. K |
≥5mg |
Cu |
≥10mg |
lysine |
≥13.2g |
|
|
Vit. B1 |
≥13mg |
Fe |
≥120mg |
methionine and cystine |
≥7.8g |
|
|
Vit. B2 |
≥12mg |
Zn |
≥30mg |
arginine |
≥11g |
|
|
Vit. B6 |
≥12mg |
Mn |
≥75mg |
tryptophan |
≥2.5g |
|
|
Vit. B12 |
≥0.022mg |
I |
≥0.5mg |
histidine |
≥5.5g |
|
|
biotin |
≥0.2mg |
Se |
0.1-0.2mg |
phenylalanine and tyrosine |
≥13.0g |
|
|
niacin |
≥60mg |
Ca |
10-18g |
threonine |
≥8.8g |
|
|
pantothenic acid |
≥24mg |
all phosphorus |
6-12g |
leucine |
≥17.6g |
|
|
folic acid |
≥6mg |
water |
≤100g |
isoleucine |
≥10.3g |
|
|
choline |
≥1250mg |
crude ash |
≤80g |
valine |
≥11.7g |
|
- Cadmium was administrated during 17 consecutive days? Including the weekends?
Response: Yes (mimicking the environment of industrial zones)
- How were these animals sacrificed?
Response: The animals were anesthetized using intraperitoneal injection of 2,2,2-tribromoethanol solution before they were sacrificed.
- During the experimental assay what biological variables were registered? Food consumption? Water consumption? Body weight variation?
Response: The variation in weight was recorded through out experiment (weight was recorded after each 4 days). While on final day, the body weigh, body length, thoracic circumference, and abdominal circumference was recorded. because of no effect recorded, these results are not included in article
- In my opinion, points 2.6 and 3.2 are not Daily Sperm Production, but Sperm Production at the end of the experimental protocol, ie, at animals' sacrifice.
Response: in material and methods, the refrence is given from where we modified the protocol/formula for our study. Similar kind of protocol is reported by Jahan et al.,2016 as Daily sperm Production.
- Figure 1 repeats the information placed in table 1. So, figure 1 should be removed.
Response: Table 1 has been edited
- Authors should place on table 1 the relative weight of testis. The relative weight of testis is calculated by dividing testis weight/animal weight.
Response: The relative mass of testis is added in figure 1 (line 160 to 163)
- How was calculated: Weight gain/loss?
Response: Animals were weighted before acclimatization to lab environment. Then the animals were weighted on day one of dosage starting. The weight gain/loss is calculated by
Weight gai/loss = final weight – initial weight(day 1 of dosage)
- What is the impact of three different doses of cadmium on animals' food and water consumption?
Response: During animal trails only variation in animal weight were recorded. In next phase of experiment these parameters will be taken into account
- Were collected blood samples? And Kidneys? What are the impacts of cadmium on kidney function: kidney histology and creatinine?
Response: the major focus of current study was on reproductive physiology. that’s why kidney and liver are not discussed in article
- What is the real impact of this trial for future work? What is its innovation?
Response: The current study was designed according to guidelines of ATSDR (Agency of Toxic Substances and Disease Registry). Speaking of oral exposure, no research group mentioned the DNA damage in heritable DNA when the exposure is oral. In most cases the damage in DNA is discussed, whereas in this study the damage in heritable DNA is discussed. The daily sperm production is a very old parameter, which is seldomly used. Usually the old parameters are ignored completely by the research community. We take the protocol of Rob et al .,1970. And used that to study the effect of Cd on sperm production. Which is studied for very few chemicals.
The primary purpose of current study was to select the best possible doses for our next pharmacological study.

Round 2
Reviewer 3 Report
The authors addressed my comments. I have no further questions.
Author Response
Thank you for your suggestions, and for improving our manuscript
Reviewer 4 Report
- There are still many grammatical problems in this paper, including the wrong use of punctuation marks.
-
Some of the results in this article are inaccurately described, such as the P value on line 168.
- The author still needs to revise the article carefully.
Author Response
- There are still many grammatical problems in this paper, including the wrong use of punctuation marks.
Response: The article is rechecked thoroughly
- Some of the results in this article are inaccurately described, such as the P value on line 168.
Response: Thanks for pointing it out. All the results are checked again.
- The author still needs to revise the article carefully.
Response: The manuscript is rechecked thoroughly
Reviewer 6 Report
Abstract:
The authors should clarify the observed results between groups.
How could this can be done: “Cd toxicity 18 must be controlled using natural sources “?
Material an methods section
In my opinion the justification of using these doses “because they did not kill the animals 84 examined in previous studies“ is not sufficient. Because following this reasoning other doses could be used, many different doses could be used.
How were animals sacrificed? This justification is not enough: “according to guidelines of Jilin University” I do not know, as well as the majority of readers, what are these guidelines. Are guidelines internationally approved.
Why were evaluated five parameters in testis? Why these parameters. Please add a bibliographic reference to support this procedure.
From where was obtained this formula: ? = (?/16) × 100 × 5 × 5.5 × 1000?
Results
Authors should explained how was calculated weight gain loss and the relative weight in material and methods section. Specially on statistical analysis section.
Why are using authors # after table and figure?
I do not agree with figure 1 legend: “Figure # 1: (A) Mean (with SEM) variation in initial and final weights, showing about 30 grams decrease in weight of 15mg/Kg animals, whereas same weight is gained in control group. (B) Mean relative mass of testis of control and experimental groups “ Figure 1 shows the weight gain and not the variations of weights.
Is missing the coloration of figures magnification of figures 2 and 3, and is missing the magnification of figure 3.
I don't understand why the authors present in table 1 the average mass of the testicles and epididymis and then afterwards present a figure with the relative testicle mass. In addition to body mass gain the authors should also put mean body mass at the beginning and end of the trial.
The authors place great emphasis on animals’ loss of body weight. I would like to know the general condition of these animals. What do the authors have to say about this? Did all the animals survive? How did the water and food consumption of these animals behave? What was the outcome of the relative mass of the liver and kidneys of these animals?
Authors should indicate on table 1 the number of animals per group.
The authors cannot extend their findings to the presence of cadmium in food and water. Because their work consisted of administering cadmium by intubation to the animals and not in their drinking water or food.
I don't understand why the authors present in table 1 the average mass of the testicles and epididymis and then afterwards present a figure with the relative testicle mass. In addition to body mass gain the authors should also put the average body mass at the beginning and end of the trial.
In several references is missing information.
Author Response
Abstract:
The authors should clarify the observed results between groups.
Response: The abstract is updated with the addition of groups names in comparisons
How could this can be done: “Cd toxicity 18 must be controlled using natural sources “?
Response: The cadmium toxicity could be controlled in 2 major ways. Firstly, by remediation of Cd using different non-edible plants, and secondly, if Cd is consumed then by controlling the damage done by Cd to different organs using herbal/traditional/natural medicine.
The primary purpose of the current study was to find the best possible dose for our next experiment. In which we want to study the protective effect of different natural medicine against Cd
Material an methods section
In my opinion the justification of using these doses “because they did not kill the animals 84 examined in previous studies“ is not sufficient. Because following this reasoning other doses could be used, many different doses could be used.
Response: According to ATSDR, the smallest possible lethal dose of Cd is 15.3mg/kg. whereas in our study the highest dose is 15mg/kg and as told earlier, in the current experiment we wanted to find a best possible dose for our pharmacological study. (explanation added in manuscript)
How were animals sacrificed? This justification is not enough: “according to guidelines of Jilin University” I do not know, as well as the majority of readers, what are these guidelines. Are guidelines internationally approved.
Response: The protocol of animal sacrifice is briefly discussed in the manuscript, whereas the links for University guidelines are added to the material and method section (which are as follow) https://sydw.jlu.edu.cn/info/1009/2196.htm
http://202.198.25.15/uhtbin/cgisirsi/x/0/0/5?searchdata1=548824{ckey}
Why were evaluated five parameters in testis? Why these parameters. Please add a bibliographic reference to support this procedure.
Response: These parameters were studied according to Jahan, S.; Iftikhar, N.; Ullah, H.; Rukh, G.; Hussain, I. Alleviative effect of quercetin on rat testis against arsenic: a histological and biochemical study. Syst. Biol. Reprod. Med. 2015, 61, 89–95.
From where was obtained this formula: ? = (?/16) × 100 × 5 × 5.5 × 1000?
Response: Jahan, S.; Rehman, S.; Ullah, H.; Munawar, A.; Ain, Q.U.; Iqbal, T. Ameliorative effect of quercetin against arsenic-induced sperm DNA damage and daily sperm production in adult male rats. Drug Chem. Toxicol. 2016, 39, 290–296.
Results
Authors should explained how was calculated weight gain loss and the relative weight in material and methods section. Specially on statistical analysis section.
Response:
|
Groups (n=4) |
Control |
5 mg/Kg |
10 mg/Kg |
15 mg/Kg |
|
Initial Weight (g) |
238.25±6.46 |
254 ± 4.97 |
252.5 ± 14,26 |
287.5 ±14.26 |
|
Final Weight (g) |
266.75 ±6.69 |
289.75 ±6.41 |
235.5 ±9.5 |
253.67 ±6.89 |
|
Final – initial |
28.5 ± 2.06 |
35.75 ±8.30 |
-13 ±12 |
-29 ± 11.55 |
The weight gain/loss was calculated by subtracting the initial weight from the final weight
Change in weight = final weight – initial weight
Why are using authors # after table and figure?
Response: # has been removed in the revised manuscript
I do not agree with figure 1 legend: “Figure # 1: (A) Mean (with SEM) variation in initial and final weights, showing about 30 grams decrease in weight of 15mg/Kg animals, whereas same weight is gained in control group. (B) Mean relative mass of testis of control and experimental groups “ Figure 1 shows the weight gain and not the variations of weights.
Response: Thank you for pointing it out. The legend has been changed
Is missing the coloration of figures magnification of figures 2 and 3, and is missing the magnification of figure 3.
Response: The color of images is corrected, and the magnification of figure 3 is added in the bracket in legend
I don't understand why the authors present in table 1 the average mass of the testicles and epididymis and then afterwards present a figure with the relative testicle mass. In addition to body mass gain the authors should also put mean body mass at the beginning and end of the trial.
Response:
|
Groups (n=4) |
Control |
5 mg/Kg |
10 mg/Kg |
15 mg/Kg |
|
Initial Weight (g) |
238.25±6.46 |
254 ± 4.97 |
252.5 ± 14,26 |
287.5 ±14.26 |
|
Final Weight (g) |
266.75 ±6.69 |
289.75 ±6.41 |
235.5 ±9.5 |
253.67 ±6.89 |
|
Final – initial |
28.5 ± 2.06 |
35.75 ±8.30 |
-13 ±12 |
-29 ± 11.55 |
The mean initial and final weight has been added in table 1. Whereas the relative mass of testis was initially added to figure 1 after a suggestion from a reviewer (along with formula) (reviewer 6). Currently, that is removed from figure 1 and added to table 1
The authors place great emphasis on animals’ loss of body weight. I would like to know the general condition of these animals. What do the authors have to say about this? Did all the animals survive? How did the water and food consumption of these animals behave? What was the outcome of the relative mass of the liver and kidneys of these animals?
Response: The weight of all the animals was checked after every 4 days. Standard environmental conditions (temperature, humidity, light/dark cycle) were maintained throughout animal trails. the animals were given free access to standard food and tap water. Initially, all the groups had 10 animals, in which 3 animals in the 10 mg/kg group and 2 animals in the 15 mg/kg group died. They are not mentioned in the manuscript, because the required standard protocols (OECD guidelines) were not considered in the experimental study. Whereas according to ATSDR the smallest lethal dose is 15.3 mg/kg of Cd.
Currently, the kidneys and liver were only collected for histology, and their weights were not recorded as the focus was on the reproductive organ.
Authors should indicate on table 1 the number of animals per group.
Response: Thank you for the suggestion, the number of animals are added in the revised manuscript
The authors cannot extend their findings to the presence of cadmium in food and water. Because their work consisted of administering cadmium by intubation to the animals and not in their drinking water or food.
Response: In the current study, we considered the absorption of Cd from the intestine. Whereas the route of food, water, and oral gavage is all through the intestine
I don't understand why the authors present in table 1 the average mass of the testicles and epididymis and then afterwards present a figure with the relative testicle mass. In addition to body mass gain the authors should also put the average body mass at the beginning and end of the trial.
Response: The relative mass of testes was added after a suggestion from a reviewer. (reviewer 6) in round 1 of review. The initial and final weights are added to the final manuscript (Table 1)
In several references is missing information.
Response: thanks for pointing it out. All the references are redone using Mendeley